# Optimal Planting Time for Summer Tomatoes (*Lycopersicon esculentum* Mill.) Cropping in Korea: Growth, Yield, and Photosynthetic Efficiency in a Semi-Closed Greenhouse

**DOI:** 10.3390/plants13152116

**Published:** 2024-07-30

**Authors:** Hyo Jun Bae, Seong-Hoon Kim, Yuseok Jeong, Sungjin Park, Kingsley Ochar, Youngsin Hong, Yun Am Seo, Baul Ko, Jeong Hyang Bae, Dong Soo Lee, Inchan Choi

**Affiliations:** 1Division of Agricultural Engineering, National Institute of Agricultural Sciences, Rural Development Administration, Jeonju 54875, Republic of Korea; bhj0711@korea.kr (H.J.B.); archha98@korea.kr (Y.J.); dbtjr2665@korea.kr (S.P.); honge159@korea.kr (Y.H.); 2Department of Horticulture Industry, Wonkwang University, Iksan 54538, Republic of Korea; dogwal@hanmail.net (B.K.); bae@wku.ac.kr (J.H.B.); 3National Agrodiversity Center, National Institute of Agricultural Sciences, Rural Development Administration, Jeonju 54875, Republic of Korea; shkim0819@korea.kr (S.-H.K.); ocharking@korea.kr (K.O.); 4Department of Data Science, Jeju National University, Jeju 63243, Republic of Korea; seoya@jejunu.ac.kr; 5National Institute of Horticultural & Herbal Science, Wanju 55365, Republic of Korea; tara0808@korea.kr

**Keywords:** cooling, light use efficiency, semi-closed greenhouse, summer cropping, tomato

## Abstract

In Korea, greenhouses are traditionally used for crop cultivation in the winter. However, due to diverse consumer demands, climate change, and advancements in agricultural technology, more farms are aiming for year-round production. Nonetheless, summer cropping poses challenges such as high temperatures, humidity from the monsoon season, and low light conditions, which make it difficult to grow crops. Therefore, this study aimed to determine the best planting time for summer tomato cultivation in a Korean semi-closed greenhouse that can be both air-conditioned and heated. The experiment was conducted in the Advanced Digital Greenhouse, built by the National Institute of Agricultural Sciences. The tomato seedlings were planted in April, May, and June 2022. Growth parameters such as stem diameter, flowering position, stem growth rate, and leaf shape index were measured, and harvesting was carried out once or twice weekly per treatment from 65 days to 265 days after planting. The light use efficiency and yield per unit area at each planting time was measured. Tomatoes planted in April showed a maximum of 42.9% higher light use efficiency for fruit production and a maximum of 33.3% higher yield. Furthermore, the growth form of the crops was closest to the reproductive growth type. Therefore, among April, May, and June, April is considered the most suitable planting time for summer cultivation, which is expected to contribute to reducing labor costs due to decreased workload and increasing farm income through increased yields. Future research should explore optimizing greenhouse microclimates and developing crop varieties tailored for summer cultivation to further enhance productivity and sustainability in year-round agricultural practices.

## 1. Introduction

In Korea, where distinct seasons—spring, summer, autumn, and winter—prevail, greenhouses serve a crucial role in agriculture, particularly for winter cultivation. They provide a warm environment essential for growing crops during cold weather. However, with changing consumer demands and challenges posed by climate change, there is a growing trend towards urban agriculture and year-round production [1]. Climate change has extended summers and introduced abnormal weather patterns, increasing demand for crops during high-temperature periods [2]. Consequently, the area dedicated to facilities for year-round cultivation is expanding annually. Crops experience reduced net photosynthesis due to excessive respiration from high temperatures and intense light, leading to poor growth caused by transpiration and stress [3]. This results in a decline in product quality and yield, necessitating active measures such as shading or cooling to mitigate these effects [4,5]. Furthermore, the current state of agriculture in Korea is characterized by a declining rural population and aging [6], leading to increased labor and management costs and worsening conditions for agricultural production. To address these issues and achieve sustainable agriculture, smart farming has emerged and is being promoted [7]. For smart agriculture, smart farm technologies are being developed for open fields, facility horticulture, and livestock farming [8]. In facility horticulture, various sensors, actuators, and environmental control systems are installed in greenhouses and connected to the internet [9]. This allows for the observation of internal conditions and the adjustment of settings to control the internal environment from anywhere, at any time. Additionally, external weather stations, internal temperature and humidity sensors, nutrient solution monitors, leaf temperature sensors, and thermal imaging cameras collect environmental data from both inside and outside, providing objective data that increases the convenience of farm operations [10]. The accumulated data analysis enables annual improvements to the internal environment, potentially leading to increased yields. 

Greenhouses are categorized into open, semi-closed, and closed types [11,12,13]. Open greenhouses often have side vents and are commonly single-span structures, while closed greenhouses are vertical indoor farms. A semi-closed greenhouse has a higher internal pressure compared to the outside and minimizes the intake of external air by using active heating and cooling systems [14]. By creating a positive pressure inside the greenhouse, the environment can be maintained uniformly and stably, reducing the need for pesticides by preventing pest infiltration [12]. Minimal ventilation with external air saves energy and maintains a high concentration of CO_2_, which can increase crop productivity [14,15,16,17]. However, currently, more than 95% of Korean greenhouses are plastic-covered, and most of the semi-closed greenhouses in Korea are glass greenhouses. Typically, tomatoes are transplanted in August and grown until June of the following year, allowing for the harvest of more than 30 flower clusters. There have been studies on seedling methods for high-temperature periods [18], optimal pruning management for high-temperature periods [19], and comparisons of paprika growth and yield in semi-closed greenhouses and regular plastic greenhouses during high-temperature periods [20]. However, research on the appropriate transplanting time for summer cropping is still insufficient. In Korea, summer cropping is challenging due to high temperatures, high humidity from monsoons, and low light conditions, leading to poor fruit set and the occurrence of misshapen fruits, which reduces productivity and makes it less favorable for farmers. Nevertheless, domestic tomato production decreases during this period. Therefore, in greenhouses equipped with cooling facilities, an increase in average prices can be expected, potentially boosting farmers’ incomes. This study aims to determine the most suitable transplanting time for summer cropping of tomatoes in April, May, or June in Korean-style semi-closed plastic greenhouses with heating and cooling capabilities. The current work investigated the growth dynamics, yield variations, and chlorophyll fluorescence responses of summer tomatoes planted at different times, aiming to provide insights into optimal strategies of tomato cultivation in summer seasons in Korea. Using an advanced greenhouse facility provides an ideal setting for investigating the growth and development of tomato plants under controlled environmental conditions as well as helping to provide reliable and reproducible data in crop experiments [21]. 

## 2. Results 

### 2.1. Growth Analysis Based on the Transplanting Time

Significant differences (*p* ≤ 0.05) were observed in stem diameter and flowering position across different planting times. Generally, tomatoes planted in June had the largest growth, while those planted in April had the smallest (Table 1 and Table 2).

#### 2.1.1. Stem Growth

The early growth investigation was conducted from 75 to 89 days after transplanting. The stem growth for tomatoes planted in April ranged from 17.1 cm to 24.7 cm, showing no significant differences from other planting times. The late growth investigation was conducted from 244 to 258 days after transplanting. The stem growth for tomatoes planted in April ranged from 9.4 cm to 12.5 cm, significantly shorter compared to those planted in June, which ranged from 18.6 cm to 21.5 cm (Table 1).

#### 2.1.2. Leaf Shape Index

A higher leaf shape index indicates longer and narrower leaves, while a lower index indicates shorter and wider leaves. The leaf shape index for tomatoes planted in April was significantly the highest, ranging from 1.14% to 1.29%. Tomatoes planted in May had a leaf shape index ranging from 0.96% to 1.19%, and those planted in June had a leaf shape index ranging from 0.96% to 1.15%. In conclusion, tomatoes planted in June showed the most vigorous growth, while those planted in April exhibited reproductive growth with less overall growth (Table 1).

#### 2.1.3. Stem Diameter and Flowering Position

Tomatoes planted in April had a stem diameter ranging from 8.9 mm to 10.2 mm and a flowering position ranging from 14.4 cm to 21.3 cm (Table 2). Tomatoes planted in May had a stem diameter ranging from 9.5 mm to 11.9 mm and a flowering position ranging from 22.1 cm to 27.6 cm, indicating a weak vegetative growth type. Tomatoes planted in June had a stem diameter ranging from 12.0 mm to 13.4 mm and a flowering position ranging from 28.0 cm to 30.5 cm, indicating strong vegetative growth.

### 2.2. Yield Variation by Planting Time

The yield per unit area (m^2^) of tomatoes was measured based on the planting time. Tomatoes planted in April yielded 12.57 kg/m^2^; those planted in May yielded 10.8 kg/m^2^; and those planted in June yielded 9.43 kg/m^2^. April-planted tomatoes had the highest yield, while those planted in June had the lowest. The weekly cumulative yield difference observed from 75 to 89 days after planting persisted until the end of the harvest period (Figure 1a). The cumulative solar radiation from the planting date to the last harvest date and the light use efficiency for fruit production (LUE_F_: g·MJ^−1^) were compared. The cumulative solar radiation for April was 3157.3 MJ/m^2^, with a LUE_F_ of 3.53 g·MJ^−1^. For May, the cumulative solar radiation was 2911.7 MJ/m^2^, with a LUE_F_ of 3.08 g·MJ^−1^. For June, the cumulative solar radiation was 2673.6 MJ/m^2^, with a LUE_F_ of 2.47 g·MJ^−1^. Additionally, the LUE_F_ for April was 3.53 g·MJ^−1^, which was 14.6% higher than May and 42.9% higher than June (Figure 1b).

### 2.3. Chlorophyll Fluorescence Response

Given that chlorophyll fluorescence is influenced by current weather conditions, the PI_ABS_, ET_0_/RC, and DI_0_/RC results measured on the same day were compared (Table 3 and Figure 2).

PI_ABS_: This index integrates the absorption capacity, electron transport efficiency, and electron trapping efficiency of PSII, reflecting overall photosynthetic activity [22]. The PI_ABS_ values for April were 3.677 during early growth and 3.468 during late growth (Table 3). For May, the values were 3.702 and 3.182, respectively, and for June, the values were 2.685 and 1.764, respectively. There was no significant difference between April and May, but June had significantly lower values, indicating reduced photosynthetic activity.

ET_0_/RC: This index reflects the electron transport efficiency of PSII, with lower values indicating higher stress [23]. The ET_0_/RC for April was 1.163 during early growth and 1.104 during late growth (Table 3). For May, the values were 1.163 and 1.124, respectively, and for June, the values were 0.999 and 0.858, respectively. There was no significant difference between April and May, but June had significantly lower values, indicating higher stress.

DI_0_/RC: This index indicates the amount of absorbed light energy that is not used for photosynthesis but is lost as heat [24]. The early growth DI_0_/RC values were 0.439 for April, 0.432 for May, and 0.444 for June, showing no significant differences (Table 3). However, during late growth, the values were 0.464 for April, 0.455 for May, and 0.637 for June, with June showing significantly higher values. 

## 3. Discussion

### 3.1. Growth Analysis Based on the Transplanting Time

Understanding the optimal planting time for tomato cultivation is crucial for maximizing growth, photosynthetic efficiency, and, consequently, fruit yield [25]. Environmental parameters such as relative humidity and CO_2_ concentration were carefully controlled to ensure stable conditions conducive to accurate observation and analysis. The analysis focused on growth differences observed among tomatoes transplanted at different times. In tomatoes, stem diameter and flowering position are important physiological traits that reflect the plant’s developmental stage and reproduction strategy and, subsequently, influence fruit setting and yield potential [26]. Significant variations in stem diameter and flowering position were evident across the planting times, indicating distinct growth patterns. Tomatoes planted in June exhibited the largest growth, characterized by thicker stems and higher positioning of flowers, indicative of strong vegetative growth. In contrast, April-planted tomatoes showed smaller stem diameters and lower flower positions, suggesting a reproductive growth type with less overall vigor. Flowering position is the distance from the epical meristem to the tip of the inflorescence. A shorter distance means that more flowers bloomed faster, which leads to more fruit, higher yields, and increased assimilation distribution to the fruit. 

### 3.2. Yield Variation by Planting Time

Yield is an indispensable trait in tomato cultivation and crucial for assessing the overall productivity and economic viability. However, tomato yield can be influenced by planting time during specific seasons of cultivation [27]. The current study also assessed tomato yield across different planting times. April-planted tomatoes yielded the highest at 12.57 kg/m^2^, followed by May-planted tomatoes at 10.8 kg/m^2^, and June-planted tomatoes at 9.43 kg/m^2^. This yield trend persisted throughout the harvesting period, indicating a consistent advantage for April-planted tomatoes despite variations in solar radiation and light use efficiency. The higher yield recorded in April-planted tomatoes can be attributed to their initiation of fruit setting before the onset of monsoon conditions, whereas June-planted tomatoes experienced lower initial fruit set due to adverse weather during their vegetative phase. It has been shown that tomato yield increases linearly with the increase in cumulative solar radiation [28,29], implying that tomato plants benefit from longer exposure to sunlight during their critical growth stages [30]. This extended exposure enhances photosynthesis, which is essential for robust vegetative growth, flower production, and, ultimately, fruit development [31]. Therefore, April-planted tomatoes, starting their fruit setting earlier in the summer season, not only avoid the detrimental effects of monsoon weather but also capitalize on a longer period of favorable solar radiation. This prolonged exposure to sunlight allows them to accumulate more energy and nutrients, translating into higher yields compared to tomatoes planted later in the season, such as in June. The summer climate in Korea, characterized by high temperatures, low light, and humid monsoons, reduces net photosynthesis and increases stress on crops. Tomatoes planted in April initiated their fruit setting before the onset of the monsoon, whereas those planted in June likely experienced low initial fruit set due to the onset of the monsoon during their vegetative growth phase, leading to reduced yield. Tomatoes planted in April yielded 1.77 kg/m^2^ more than those planted in May and 3.14 kg/m^2^ more than those planted in June. However, tomatoes planted in April had a stem diameter ranging from 8.9 mm to 10.2 mm lower than the diameters recorded for May and June. These results suggest that while April-planted tomatoes yield higher due to their early fruit setting and extended exposure to beneficial solar radiation, they may exhibit comparatively thinner stems. The thinner stem diameter observed in April-planted tomatoes could indicate potential trade-offs between structural integrity and reproductive output, influencing their overall resilience and support needs throughout their growth cycle. 

### 3.3. Chlorophyll Fluorescence Response

A comprehensive understanding of tomato physiological responses can also be established based on chlorophyll fluorescence measurements [22,32]. The absorbed light in plants is utilized in various ways. When light is absorbed, electrons are excited to a higher energy level within plant cells [33]. This elevated energy level must be reduced quickly to prevent cellular stress. Plants dissipate their absorbed light energy in three primary ways: through photosynthesis, emission as fluorescence by photosystem II (PSII), or as heat [34,35]. Increased stress reduces the efficiency of photosynthesis and fluorescence emission, resulting in more energy being dissipated as heat, which raises leaf temperature [35]. In this study, chlorophyll fluorescence measurements provided insights into the photosynthetic activity and stress responses of tomatoes under different planting times. The photosynthetic performance, indicated by PI_ABS_ values, was highest for April-planted tomatoes and significantly lower for June-planted tomatoes during both the early and late growth stages. Similarly, indices reflecting electron transport efficiency (ET_0_/RC) and energy dissipation as heat (DI_0_/RC) confirmed higher stress levels in June-planted tomatoes, particularly during the late growth stages. These results emphasize the sensitivity of tomato physiology to planting time, with April having been found as the optimal time for maximizing photosynthetic efficiency and minimizing stress-induced energy loss. This further indicates that more energy was dissipated as heat in June-planted tomatoes, reflecting higher stress and less efficient photosynthesis. These results reveal the importance of planting time on the growth, yield, and physiological stress responses of tomatoes in the semi-closed greenhouse environment. Thus, planting tomatoes in April results in the most favorable growth and yield outcomes, while planting in June leads to increased stress and reduced productivity. The findings in this study contribute valuable data to the field of agricultural science, providing a basis for informed decision-making by farmers and researchers aiming to enhance crop productivity and sustainability in greenhouse cultivation systems. By elucidating the impact of planting time on growth, photosynthetic efficiency, and yield, this study also emphasizes the importance of strategic planting practices tailored to local environmental conditions, thereby optimizing tomato production.

## 4. Materials and Methods

### 4.1. Experimental Site and Greenhouse Design

The experiment was conducted from April to August in 2022 using an advanced digital greenhouse established in 2021 by the National Institute of Agricultural Sciences in Iseo-myeon, Wanju-gun, Jeollabuk-do (32°39′40″ N, 51°40′49″ E) (Figure 3). It is a semi-closed Venlo-type greenhouse structure, and the covering material used is PO film. The entire greenhouse has a height of 8.3 m, a side height of 7.0 m, a width of 40 m, and a length of 120 m, consisting of five sections (24 m × 40 m each). Each greenhouse can individually control temperature, humidity, solar radiation, CO_2_, nutrient supply, heating, cooling, and actuators. For heating and cooling, the greenhouse uses six 840 kW air heat pumps (ACHH040LET2, LG, Republic of Korea), which store thermal and cooling energy in a 300-ton thermal storage tank. This energy is then distributed using tube rails and Fan Coil Units (FCUs). Additionally, the air conditioning room is equipped with internal and external air conditioning windows and FCUs, which selectively intake external air based on the internal and external temperatures. The irrigation method utilized solar radiation proportional control and was supplied at intervals of 80 J/m^2^ to 120 J/m^2^. The amount supplied per time was 100 cc/plant to 150 cc/plant.

### 4.2. Plant Material and Treatment Conditions

The tomatoes transplanted were grafted seedlings, using ‘Dafnis’ (Syngenta Korea, Seoul, Republic of Korea) as the scion and ‘Spider’ (Takii Korea, Seoul, Republic of Korea) as the rootstock. The coir substrate (Daeyoung GS, Republic of Korea) used measured 1 m in length, 0.2 m in width, and 0.1 m in height, with a chip-to-dust ratio of 7:3. The substrate was moistened to an EC concentration of 2.5 dS/m. The greenhouse was divided into three sections at 8 m intervals, and 6-week-old tomato plants were transplanted on 20 April, 19 May, and 21 June, with 360 plants in each section. The planting density was 1.13 plants per square meter, with a bed spacing of 1.6 m. The target drainage EC was set to 3.5–4.0 dS/m; the pH was set to 5.5–6.0; and the drainage rate was set to 35–40%. Based on changes in the drainage EC and pH, the nutrient solution EC was maintained at 2.3–2.5 dS/m and the pH at 5.5–6.0.

### 4.3. Environmental Conditions during the Experiment

During the experimental period, the monthly average external temperature ranged from −0.3 °C to 28.0 °C, and the monthly cumulative solar radiation ranged from 176.7 MJ/m^2^ to 532.4 MJ/m^2^. The internal temperature of the greenhouse was maintained between 17.1 °C and 23.9 °C. The relative humidity ranged from 65.9% to 85.1%, and the CO_2_ concentration was controlled between 367 ppm and 520 ppm (Figure 4).

### 4.4. Growth Assessment and Yield Measurement

To evaluate the growth and yield of tomatoes, 360 tomato plants were transplanted in each experimental stage, of which 36 plants were selected from the center for measurement. Growth comparisons were made during the early and late stages of growth over a three-week period. Early growth was evaluated from 75 to 89 days after transplanting, while late growth was assessed from 244 to 258 days after transplanting. The parameters measured included the stem diameter, flowering position, stem growth, leaf length, and leaf width. The leaf shape index was calculated by dividing the leaf length by the leaf width. Flowering position was determined by measuring the distance between the highest fully bloomed flower cluster and the growing point. Stem diameter was measured just below the flowering cluster. Stem growth was determined by measuring the distance from the mark on the support string, indicating the position of the growing point 7 days prior to the growing point. Leaf length and width were measured by assessing the horizontal and vertical dimensions of the lower leaves of the uppermost flowering cluster. Tomato fruit harvesting was conducted from 65 to 265 days after transplanting for each treatment, at intervals of 1 to 2 times per week. Fruits were harvested when they were more than 80% colored (red stage) and weighed using an electronic scale (KS-308, Dretec, Japan).

### 4.5. Chlorophyll Fluorescence Measurement

Light is closely related to crop yield, so chlorophyll fluorescence measurements were taken to compare the light use efficiency of crops planted at different times. Chlorophyll fluorescence was measured three times each during the early and late growth stages. The early growth measurements were taken at 95, 124, and 152 days after transplanting, while the late growth stage measurements were taken at 208, 237, and 264 days after transplanting. Measurements were conducted on the terminal leaves of the lower leaf of the highest flowering cluster for 36 plants in each treatment group. The leaves were dark-adapted for 30 min using a leaf clip, and chlorophyll fluorescence was measured using a Fluorpen FP-100 (Photon Systems Instruments, Brno, Czech Republic) to conduct the Fast chlorophyll a fluorescence induction (OJIP) test.

The OJIP test captures the following fluorescence signals:O phase: The minimum fluorescence measured at 50 µs when all photosystem II (PSII) reaction centers are open.J phase: Measured at 2 ms.I phase: Measured at 60 ms.P phase: The maximum fluorescence (F_M_) measured when all PSII reaction centers are closed [24].

The data from the OJIP test were analyzed using FluorPen software (Photon Systems Instruments, 1.1.2.4 Version, Brno, Czech Republic). The indices derived from the OJIP test include F_0_, F_M_, F_V_/F_M_, PI_ABS_, DI_0_/RC, and ET_0_/RC and are defined in Table 4.

F_0_: Indicates the energy emitted by chlorophyll molecules before being transferred to the PSII reaction center. An increase in F_0_ signifies a greater number of inactive chlorophyll molecules, indicating reduced energy capture capacity [15].F_V_/F_M_: The most commonly used index in chlorophyll fluorescence analysis, representing the maximum photochemical efficiency of PSII [36]. It is calculated using the following formula: F_V_ = F_M_ − F_0_.PI_ABS_: An integrated index representing the absorption, electron transport efficiency, and electron trapping efficiency of PSII, reflecting overall photosynthetic activity [22,32].ET_0_/RC: Reflects the electron transport efficiency of PSII, specifically the reduction of QA to QB. This index decreases under stress conditions [23].DI_0_/RC: Indicates heat loss per reaction center (RC) at time zero. Energy not used for photosynthesis or emitted as fluorescence is dissipated as heat, which increases under stress [37].

### 4.6. Statistical Analysis

To compare the growth parameters, including the stem diameter, flowering position, stem growth, and leaf shape index, of tomatoes transplanted at different times in the semi-closed greenhouse, a statistical analysis was performed. The analysis was conducted using SPSS v.27 (IBM Corporation, Chicago, IL, USA). Duncan’s Multiple Range Test (DMRT) was used to determine significant differences between treatments, with the confidence level set at 95%.

## 5. Conclusions

The results showed that the tomatoes planted in April had a stem diameter approximately 18.4% thinner than those planted in May and 34.7% thinner than those planted in June. Additionally, the flowering position of tomatoes planted in April was 66.7% higher than those planted in May and 44.0% higher than those planted in June. Moreover, stem growth in tomatoes planted in April was about 29.0% shorter compared to those planted in May and 34.6% shorter compared to June. Reduced stem growth suggests a decrease in the frequency of stem lowering and guiding operations, potentially reducing labor costs. The leaf shape index for April plantings was about 11.9% higher compared to May and 14.6% higher compared to June, indicating a more balanced and weaker reproductive growth type. The cumulative solar radiation in April was 8.4% higher than in May and 18.1% higher than in June. The cumulative yield of tomatoes planted in April was 16.4% higher than those planted in May and 33.3% higher than those planted in June. Additionally, the light use efficiency for fruit production (LUE_F_) in April was 14.6% higher than in May and 42.9% higher than in June. Therefore, April was identified as the period with the highest cumulative solar radiation and light use efficiency. Chlorophyll fluorescence measurements showed no significant differences between April and May. However, the PI_ABS_ values for June were 36.9% lower during early growth and 96.6% lower during late growth compared to April, indicating the lowest photosynthetic activity in June. ET_0_/RC values for June were 16.4% lower during early growth and 28.7% lower during late growth compared to April. The DI_0_/RC values for June showed no significant differences during early growth but were 37.3% higher during late growth compared to April and 40.0% higher compared to May, indicating higher stress levels and greater heat loss in June-planted tomatoes. Our analysis of growth, yield, and photosynthesis determined that April is the most suitable planting time for summer cropping in Korea. Planting in April can reduce labor costs and increase farm income due to higher yields. Future research should focus on the effects of supplementary lighting during the summer monsoon season in Korea, as this could further optimize growth conditions. Since this study was conducted on “Dafnis” and “Spider” grafted seedlings, results may vary for other varieties, highlighting the need for continued investigation across different tomato cultivars.

## Figures and Tables

**Figure 1 plants-13-02116-f001:**
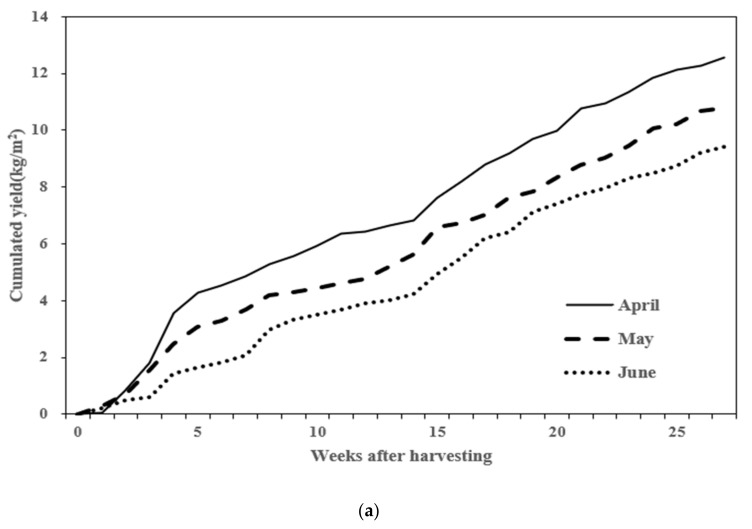
Comparison of harvest yield by planting period. (**a**) Cumulative harvest yield; (**b**) light use efficiency for fruit production (LUE_F_).

**Figure 2 plants-13-02116-f002:**
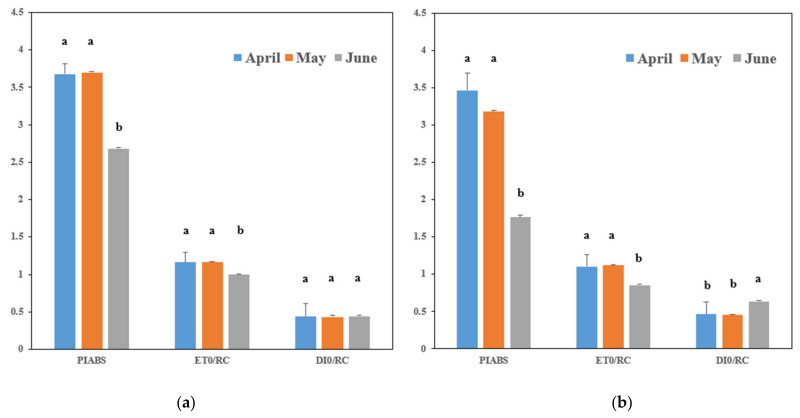
Chlorophyll fluorescence measurements according to tomato planting times. (**a**) Measurements during the early growth stage; (**b**): measurements during the late growth stage. The letters a and b indicate the presence or absence of significant variability. Bars with the same letters within each group signify no significant difference, while different letters indicate presence of significant variability (*p* < 0.05) among the groups.

**Figure 3 plants-13-02116-f003:**
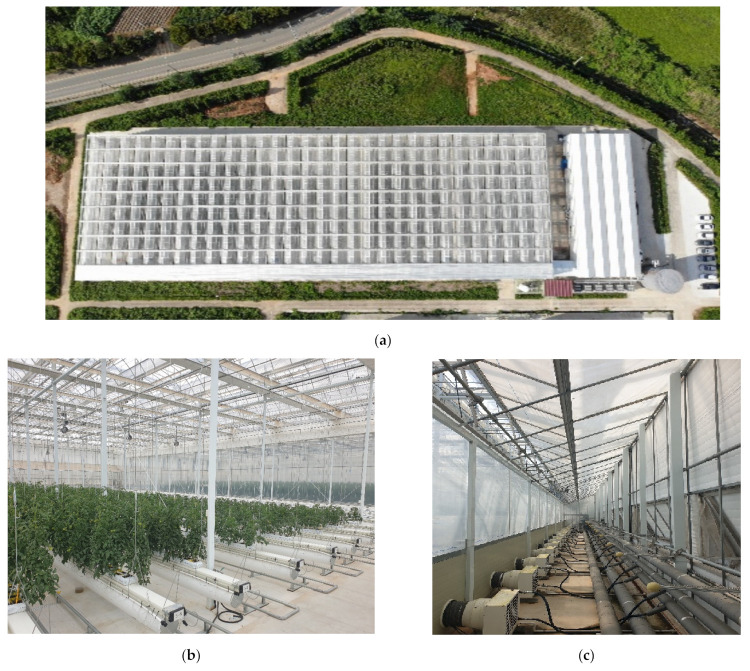
Overview and interior of the advanced digital greenhouse. (**a**) An overview of the advanced digital greenhouse; (**b**) the interior of the greenhouse; and the (**c**) air conditioning room.

**Figure 4 plants-13-02116-f004:**
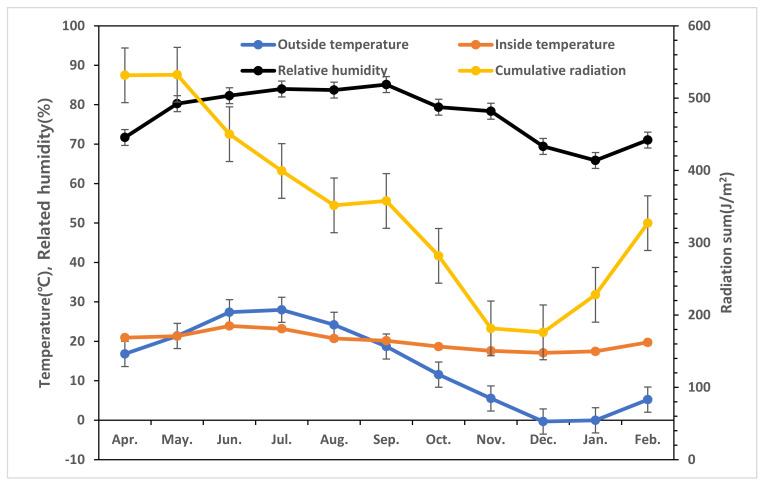
Average outside temperature (●), internal temperature (●), relative humidity (●), and total radiation (●) during the period of cultivation of tomatoes grown in greenhouses located in Jeonju.

**Table 1 plants-13-02116-t001:** Plant length and leaf shape index of tomato plants according to the planting season.

	Planting Season	Days after Planting (DAP)					
		Early Growth			Late Growth		
		75	82	89	244	251	258
Stem Growth ^z^ (cm/week)	April	17.1 c	21.6 b	24.7 a	10.6 c	9.4 c	12.5 c
	May	23.9 a	28.3 a	25.8 a	15.5 b	15.3 b	15.0 b
	June	21.8 b	21.5 b	25.8 a	18.6 a	20.1 a	21.5 a
	LSD (5%) ^y^	***	**	ns	***	***	***
Leaf shape index ^z^ (%)	April	1.16 a	1.14 a	1.22 a	1.22 a	1.23 a	1.29 a
	May	1.19 a	1.09 ab	1.08 b	1.07 b	1.01 c	0.96 c
	June	0.98 b	1.01 b	0.96 c	1.05 b	1.15 b	1.08 b
	LSD (5%) ^y^	**	*	***	**	***	***

^z^ The stem growth was measured by determining the distance from the guide string marked at the position of the growing point (epical meristem) 7 days prior to the current growing point. Leaf shape index = leaf length/leaf width. ^y^ Mean separation within columns by Duncan’s multiple range test at *p* = 0.05. *, **, and ***: Significant at *p* ≤ 0.05, 0.01, and 0.001, respectively. The letters a, b, and c indicate the presence or absence of significant variability. The same letters shared between groups signify no significant differences, while different letters indicate significant variability (*p* < 0.05) among the groups.

**Table 2 plants-13-02116-t002:** Stem diameter and flowering position of tomato plants according to the planting season.

	Planting Season	Days after Planting (DAP)					
		Early Growth			Late Growth		
		75	82	89	244	251	258
Stem diameter (mm) ^z^	April	9.0 c	8.9 b	9.1 b	10.2 b	8.7 b	9.3 b
	May	11.6 b	9.5 b	11.9 a	12.6 a	11.5 a	8.1 b
	June	13.4 a	12.8 a	12.0 a	13.1 a	11.0 a	12.0 a
	LSD (5%) ^y^	***	**	**	**	**	**
Flowering position ^z^ (cm)	April	14.4 c	16.8 c	15.6 b	21.3 b	15.1 c	16.1 b
	May	24.6 b	22.1 b	27.6 a	26.5 a	30.6 a	34.1 a
	June	30.5 a	30.3 a	28.0 a	19.9 b	19.9 b	18.8 b
	LSD (5%) ^y^	***	***	**	**	***	**

^z^ The flowering position was expressed as the distance between the flowering truss and the head of the plant. The stem diameter was measured just below the topmost flowering cluster. ^y^ Mean separation within columns by Duncan’s multiple range test at *p* = 0.05. **, and ***: significant at *p* ≤ 0.01, and 0.001, respectively. The letters a, b, and c indicate the presence or absence of significant variability. The same letters shared between groups signify no significant differences, while different letters indicate significant variability (*p* < 0.05) among the groups.

**Table 3 plants-13-02116-t003:** Results of chlorophyll fluorescence measurement according to planting period.

	Planting Season	Days after Planting					
		Early Growth			Late Growth		
		96	124	152	209	237	265
PI_ABS_	April	2.978 b	4.198 a	3.677 a	4.661 a	3.030 a	3.468 c
	May	5.274 a	3.702 b	2.423 b	3.277 b	3.182 a	4.135 b
	June	2.685 b	2.175 c	3.450 a	1.764 c	2.183 b	5.012 a
	LSD (5%) ^y^	**	***	**	***	**	***
ET_0_/RC	April	1.044 b	1.052 b	1.163 a	1.047 a	0.815 b	1.104 a
	May	1.140 a	1.163 a	0.803 c	0.790 c	1.124 a	1.069 a
	June	0.999 c	0.753 c	0.888 b	0.858 b	0.813 b	1.087 a
	LSD (5%) ^y^	***	***	***	***	**	Ns
DI_0_/RC	April	0.427 b	0.371 b	0.439 a	0.354 b	0.350 b	0.464 a
	May	0.358 c	0.432 a	0.428 a	0.308 b	0.455 a	0.375 b
	June	0.444 a	0.418 a	0.353 b	0.637 a	0.456 a	0.350 b
	LSD (5%) ^y^	***	**	**	**	**	**
			Investigated: 1 August 2022			Investigated: 1 December 2022	

^y^ Mean separation within columns by Duncan’s multiple range test at *p* = 0.05. **, and ***: significant at *p* ≤ 0.01, and 0.001, respectively. The letters a, b, and c indicate the presence or absence of significant variability. The same letters shared between groups signify no significant differences, while different letters indicate significant variability (*p* < 0.05) among the groups.

**Table 4 plants-13-02116-t004:** Description of OJIP parameters.

OJIP Parameters	Explanation
F_0_	Minimal fluorescence yield of dark-adapted PS II
F_M_	Maximal fluorescence yield of dark-adapted PS II
F_V_	Maximal variable fluorescence (F_V_ = F_M_ − F_0_)
PI_ABS_	Performance index for energy conservation from photons absorbed by PSII antennas to the reduction of Q_B_
F_V_/F_M_	Maximum quantum yield of primary PSII photochemistry
ET_0_/RC	Electron transport (ET) flux from Q_A_ to Q_B_ per PS II reaction center (RC)
DI_0_/RC	Heat dissipation (DI) at time zero per reaction center

## Data Availability

All data are available within the paper.

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
