# Peer review of "Optimal Planting Time for Summer Tomatoes (Lycopersicon esculentum Mill.) Cropping in Korea: Growth, Yield, and Photosynthetic Efficiency in a Semi-Closed Greenhouse"

_plants, 2024, doi:10.3390/plants13152116_

Round 1

Reviewer 1 Report

Comments and Suggestions for Authors

DearAuthors,
In my opinion, the manuscript presented for the assessment, entitled: "Optimal Planting Time for Summer Tomato Cropping in Korea: Growth, Yield, and Photosynthetic Efficiency in a Semi-Closed Greenhouse” is generally written in the correct form and the Authors presented a wide range of diverse results. The information contained in it could be cognitive and application significant and I think the presented data and conclusions could interest many researchers and readers after some refilling. Although the work is very interesting and of application value I think that the Authors should take into account some modification of this article. I recommend publishing it in "Plants” after a minor revision.

General comments: the weak point of this manuscript is the literature section. The works cited here are too often items from post-conference materials, I propose to enrich this manuscript with more items regarding the discussed topic and contemporary items.

Particular comments:

Title

Lines 2-4: "Optimal Planting Time for Summer Tomato Cropping in Korea: Growth, Yield, and Photosynthetic Efficiency in a Semi-Closed Greenhouse”-

1.      “Summer Tomato” is a common name, the Latin species name should be used in the title or at the beginning at the text.

2.      there is no clear connection between optimal planting time and chosen by Authors plant parameters; should be clarified

Abstract

Line 16: However due to diverse consumer demands …… - cancel "however"

Keywords

Line 33: should be in alphabetical order and I suggest adding some keywords of botanical meaning, for example, use the name of the plant

Introduction

Line 36 – 40: “In Korea, there are four distinct seasons: spring, summer, autumn, and winter. Korean greenhouses are primarily constructed for the purpose of cultivating and harvesting crops during the winter. Therefore, greenhouses in Korea imply a warm space. However, due to the diverse demands of consumers, urban agriculture and the number of farms aiming for year-round production is increasing” - this information does not follow from one another and is written in an awkward style; should be clarified

Line 84 - 85: “For this purpose, the growth and yield 84 of the crops were compared and analyzed.” – according to the title, the photosynthetic activity should be taken into account also. and photosynthetic activity

In addition, there is no information in the introduction about what plant parameters should be taken into account, or which the Authors chose, believing that they would be an appropriate indicator in their research assumptions.

Results

Line 88 - 95: “In this study, a total of 360 tomato plants were carefully transplanted into a state-of the-art digital greenhouse situated at the National Institute of Agricultural Sciences in  Jeonju. Throughout the course of the experiment, the relative humidity and the concentration of CO2 were regulated and rigorously monitored to ensure stable environmental conditions essential for accurate observation and analysis of plant responses. To analyze the growth differences based on planting time, 36 plants from each treatment, totaling 108 plants, were examined for flowering position, stem diameter, stem growth, and leaf shape  index (LSI) at both early and late growth stages.” - in my opinion this part suits to Materials and Methods chapter and should be canceled from here.

Line 95: “The standard  …. “ - what do you mean "standard', standard for what age and growth conditions for example?

Line 97 - 99: “Flowering position is judged based on a standard of 15cm to 20cm; shorter distances indicate reproductive growth, and longer distances  indicate vegetative growth [21].” - citations are not used in the "Results" section, should be transferred into the Introduction or/and into the discussion section

Line 131: “Table 2. Stem diameter and Flowering position of tomato plants according to planting season.” – “Flowering” from a small letter

Line 132: “z The flowering position was expressed as the distance between the flowering truss and the head of the plant, Stem …. “ – “Stem” from a small letter 

Discussion

Line 185 – 191: “The current work investigated the growth dynamics, yield variations, and chlorophyll fluorescence responses of summer tomatoes planted at different times, aiming to provide insights into optimal strategies of tomato cultivation in summer seasons in Korea. Using advanced 188 greenhouse facility provides an ideal setting for investigating the growth and development of the tomato plants under controlled environmental conditions as well as helping to provide reliable and reproducible data in crop experiments [26]. – this part suits better to Introduction section, in my opinion should be transferred there.

Materials and Methods

Line 279 – 280: “The tomatoes transplanted were grafted seedlings, using ‘Dafnis’ (Syngenta Korea, 279 Seoul, Korea) as the scion and ‘Spider’ (Takii Korea, Seoul, Korea) as the rootstock.” - is it possible to determine the age or development stage of such seedlings?

Line 293: “The internal temperature was of the greenhouse was maintained …” – “was” please cancel

Line 309: it is unclear what the Authors mean by "growth point" term?

Line 311, 312: does "growing tip" mean stem apical meristem?

Line 313: does "tomato harvesting" mean fruit harvesting or whole biomass of plants; This should be clarified.

Conclusions

Line 361 – 363: “To determine the most suitable planting time for summer cropping in Korea, toma-361 toes were cultivated in a semi-closed plastic greenhouse with cooling capabilities from 362 April 20, 2022, to March 8, 2023.” -  such an introduction in the "Conclusions" section is not necessary, and should be canceled.

Line 366: “Stem growth was about 29.0% shorter compared …” – slower? Or “stem height was about 29.0% shorter compared ….?,; should be clarified.

With best regards!

Comments on the Quality of English Language

Dear Authors,

in my opinion, the manuscript's English language is good, but it needs slight correction.

Reviewer 2 Report

Comments and Suggestions for Authors

This paper presents a study on determining the most suitable planting time for summer cropping of tomatoes in Korea. Tomatoes were cultivated in a semi-closed plastic greenhouse with cooling capabilities from April 20, 2022, to March 8, 2023. The study assessed various growth parameters and yield outcomes based on planting dates in April, May, and June. Results indicated significant differences in morphology, cumulative yield, and light use efficiency for fruit production among different planting times. April planting emerged as the optimal choice, showing enhanced photosynthetic activity compared to May and June. While the results are promising to improve the yield for future reference, there is still room for improvement to explain the correlation between the plant morphology and enhanced photosynthetic capabilities.

1.      While a better fruit yield was demonstrated for April, it is not consistent with the larger stem size for the tomato plants planted in April. Can the author elaborate more on the correlation?

2.      In section 2.1.3, the authors stated that ‘Tomatoes planted in April had a stem diameter ranging from 8.9mm to 10.2mm and a flowering position ranging from 14.4cm to 21.3cm, indicating a weak reproductive growth type’. How is flowering position related to the reproductive growth type for tomato plants? Flowering position can have both advantages and disadvantages for plant reproduction. It seems difficult to draw a conclusion based on the observation. Can the authors be more specific about the criterion for vegetative growth and reproductive growth?

3.      In Tables 1 and 2, the authors used a, b, and c to show the significant difference within a column. It would help the reader to understand if the authors specify the meaning of a,b,c in Table 1 and what the significance is based on.

4.      In line 231, the author mentioned ‘When light is absorbed, the energy level within plant cells increases [34].’ This is not a scientifically accurate way to put it. Electrons are excited to a higher energy level when light is absorbed. 

Reviewer 3 Report

Comments and Suggestions for Authors

The Manuscript entitled ‘’ Optimal Planting Time for Summer Tomato Cropping in Korea: Growth, Yield, and Photosynthetic Efficiency in a Semi-Closed Greenhouse ‘’  investigate the most suitable planting time for summer tomato cultivation in Korea. Summer cropping is challenging due to high temperatures, so the idea of the paper is interesting. Also, the structure of the paper is well. Authors provided an interesting result. However, introduction and discussion need to expand. Also, there are some comments, please see them as below:

There is some punctuation and space missing in the text. Moderate English editing is required.

Abstract

Line 23-24: add the treatments (April, May, June)! Also add the year of research.

Please add the suggestion for future research at the end of the abstract section.

Introduction

The introduction is well written, but there isn’t any comparison between current research with previous ones! Bring similar research and highlight the gap of knowledge, then compare them with the current research and explain how your work can fill these gaps. One more thing: yet, considering that tomatoes are a specific crop grown in both open-field (farm) and greenhouse environments, it would be beneficial to include some information on the disadvantages of open-field farming. So, I would suggest that in two or three sentences explain the advantages and disadvantages of these two cultivation systems. Here is a published work that you can use it to improve this gap. https://doi.org/10.3390/agronomy13030916

Results

These results sections are well written, the results are impressive, authors provided figures in appropriate resolution.

 Line 105: One space is needed between number and letter, please correct it in whole of the text.

·         ---- 17.1cm …. -- > 17.1 cm

Line 131: What’s the meaning of flowering position?!

Discussion

Like introduction, there is a lack of comparison between your work with previous research. I would suggest that the authors divide the discussion section into three parts, like the results section.

·         3.1. Growth Analysis based on Transplanting Time

·         3.2. Yield Variation by Planting Time

·         3.3. Chlorophyll Fluorescence Response

To interpretation of your results, you need to compare them with other studies. What is the strength or weakness of your results? What is your interpretation for them?

Material and methods

4.1. Experimental Site and Greenhouse Design: add the longitude and latitude of the research area.

Add the irrigation information in the material and methods section!

Conclusion

This section is well written. Add the limitation of your work in one or three lines.

References

The authors used appropriate and update references.

Comments on the Quality of English Language

 Moderate editing of English language required
